# Terminal Ileitis as the Exclusive Manifestation of COVID-19 in Children

**DOI:** 10.3390/microorganisms12071377

**Published:** 2024-07-06

**Authors:** Lea Maria Schuler, Barbara Falkensammer, Peter Orlik, Michael Auckenthaler, Christof Kranewitter, David Bante, Dorothee von Laer, Franz-Martin Fink

**Affiliations:** 1Department of Pediatrics and Adolescent Medicine, Regional Hospital St. Johann in Tirol, 6380 St. Johann in Tirol, Austria; lea.schuler@khsj.at (L.M.S.); orlik@khsj.at (P.O.); auckenthaler@khsj.at (M.A.); fink@khsj.at (F.-M.F.); 2Institute of Virology, Medical University of Innsbruck, 6020 Innsbruck, Austria; david.bante@student.i-med.ac.at (D.B.); dorothee.von-laer@i-med.ac.at (D.v.L.); 3Department of Radiology, Regional Hospital St. Johann in Tirol, 6380 St. Johann in Tirol, Austria; christof.kranewitter@khsj.at

**Keywords:** terminal ileitis, COVID-19, children, SARS-CoV-2, nucleic acid amplification, Oxford nanopore technology

## Abstract

The clinical presentation, organ involvement, and severity of disease caused by SARS-CoV-2 are highly variable, ranging from asymptomatic or mild infection to respiratory or multi-organ failure and, in children and young adults, the life-threatening multisystemic inflammatory disease (MIS-C). SARS-CoV-2 enters cells via the angiotensin-converting enzyme-2 receptor (ACE-2), which is expressed on the cell surfaces of all organ systems, including the gastrointestinal tract. GI manifestations have a high prevalence in children with COVID-19. However, isolated terminal ileitis without other manifestations of COVID-19 is rare. In March 2023, two previously healthy boys (aged 16 months and 9 years) without respiratory symptoms presented with fever and diarrhea, elevated C-reactive protein levels, and low procalcitonin levels. Imaging studies revealed marked terminal ileitis in both cases. SARS-CoV-2 (Omicron XBB.1.9 and XBB.1.5 variants) was detected by nucleic acid amplification in throat and stool samples. Both patients recovered fast with supportive measures only. A differential diagnosis of acute abdominal pain includes enterocolitis, mesenteric lymphadenitis, appendicitis, and more. During SARS-CoV-2 epidemics, this virus alone may be responsible for inflammation of the terminal ileum, as demonstrated. Coinfection with Campylobacter jejuni in one of our patients demonstrates the importance of a complete microbiological workup.

## 1. Introduction

Coronavirus disease 2019 (COVID-19) is an infectious disease caused by severe acute respiratory syndrome coronavirus type 2 (SARS-CoV-2) and exhibits a wide and nonspecific spectrum of symptoms ranging from asymptomatic infection to severe illness. In January 2020, the World Health Organization (WHO) declared this disease a public health emergency of international concern [1]. In most children, it primarily leads to mild upper respiratory tract symptoms [2,3]. Gastrointestinal symptoms such as abdominal pain, diarrhea, and vomiting are also common [4,5,6,7,8]. 

Neurological manifestations like headaches, fatigue, and confusion have been observed. A significant but rare concern is the multisystem inflammatory syndrome in children (MIS-C), also known as pediatric inflammatory multisystem syndrome (PIMS), which can lead to severe inflammation in various organs, including the heart, kidney, and gastrointestinal tract [9,10]. Cardiovascular manifestations, such as myocarditis and arrhythmias, have also been reported. Some children exhibit a skin rash, including erythematous, urticarial, and vesicular lesions. Similar to adults, children can experience persisting symptoms post-infection, known as “long COVID“, which may include fatigue, muscle and joint pain, and cognitive difficulties [11,12].

Terminal ileitis refers to inflammation of the terminal ileum. It may manifest as an acute or chronic disease. The most common chronic form is Crohn’s disease, a type of inflammatory bowel disease (IBD) causing chronic granulomatous inflammation of the terminal ileum [13]. The existence of backwash ileitis, the extension of inflammation from the colon into the terminal ileum in patients with ulcerative colitis, is discussed controversially [14]. In addition, IBD may accompany spondyloarthropathies, vasculitides, and the use of non-steroidal anti-inflammatory drugs (NSAIDs) [15,16,17]. Most cases of acute terminal ileitis are caused by infections, particularly due to Yersinia spp., Salmonella spp., Campylobacter jejuni, or Clostridium difficile. Rarely, Mycobacteriae, Actinomyces, and Histoplasma capsulatum can cause terminal ileitis, as well as Cytomegalo-, Epstein–Barr, Rota-, and Adenoviruses, and likely any virus associated with gastroenteritis [15,16,18]. Nematodes, namely Anisakis infestation, may cause terminal ileitis after raw fish consumption [19].

This case report discusses a child aged nine years in whom a SARS-CoV-2 infection manifested exclusively as terminal ileitis.

Additionally, a second case report involving a 16-month-old child with ileocecal involvement in the context of coinfection with SARS-CoV-2 and Campylobacter jejuni highlights the importance of a complete microbiological workup.

## 2. Case 1

### 2.1. History

A previously healthy 9-year-old boy with lower abdominal pain was admitted by a general practitioner who suspected an acute abdomen. The patient had experienced fever and diarrhea for three days prior to admission, followed by the development of lower abdominal pain.

### 2.2. Clinical Findings

Upon admission, the patient had a moderate fever (38.7 °C). The abdomen was diffusely tender and there was mild guarding. Apart from a positive Rovsing sign, there were no other clinical signs of appendicitis.

### 2.3. Diagnostic Evaluation

The laboratory results showed an elevated CRP level and erythrocyte sedimentation rate (CRP 9.9 mg/dL, normal range from 0.0 to 0.5 mg/dL; ESR 36 mm/h, normal range from 1 to 8 mm/h). Procalcitonin was only slightly elevated at 0.7 ng/mL (normal range from 0 to 0.2 ng/mL). Blood counts, differential counts, as well as venous blood gas analysis were normal, but mild hypoelectrolytemia was documented (serum sodium 130 mmol/L, normal range from 136 to 145 mmol/L; chloride 93 mmol/L, normal range from 98 to 107 mmol/L). Immunoglobulin-A and immunoglobulin-M levels were elevated (IgA 249 mg/dL, normal range from 53 to 204 mg/dL; IgM 326 mg/dL, normal range from 31 to 179 mg/dL), while IgG was within normal limits (991 mg/dL, normal range from 698 to 1560 mg/dL). Serum electrophoresis demonstrated low serum albumin levels (3.04 g/L, normal range from 4.02 to 4.76 g/L) and a typical acute phase reaction with elevated alpha-1-globulin (0.56 g/L, normal range from 0.21 to 0.35 g/L) and alpha-2-globulin (1.18 g/L, normal range from 0.51 to 0.85 g/L) levels.

SARS-CoV-2 was detected in throat swabs using PCR (cycle threshold of 24), identifying the XBB.1.5 Omicron subvariant. Whole-genome sequencing was performed using the Midnight Protocol by Oxford Nanopore Technologies (Oxford, UK), adapted from Freed N. et al. [20] as previously described [21]. In short, nucleic acid was extracted using the NucliSENSE Kit with the EasyMag platform (bioMérieux, Marcy l’Etoile, France). Sequencing libraries were prepared using the Midnight RT PCR Expansion Kit EXP-MRT001 and the Rapid Barcoding Kit SQK-RBK110.96 from Oxford Nanopore Technologies (Oxford, UK), and sequenced on an R9.4.1 flow cell in a MinION Mk1b sequencer according to the manufacturer’s instructions. Data processing was achieved through base calling with Guppy 6.5.7 in super high accuracy mode and passing resultant data to the epi2me-labs/wf-artic nextflow workflow [22,23]. A stool sample also tested positive for SARS-CoV-2. Further stool microbiological diagnostics were negative (negative cultures for Salmonella, Shigella, Campylobacter, and Yersinia; negative microscopic screening for enteric parasites; and negative PCR tests for Noro-, Adeno-, Astro-, Rota-, and Sapoviruses) (Seegene, Inc., Seoul, Republic of Korea). To exclude other infectious diseases, a respiratory panel PCR was also performed on the throat swab, covering a broad spectrum of pathogens. The FTD Respiratory Pathogen Assay 21 was used to screen for Adenoviruses, Influenza viruses, Parainfluenza viruses, human Metapneumovirus, human coronaviruses, Respiratory Syncytial virus, Bocaviruses, Rhinoviruses, Enteroviruses, human Parechovirus, and Mycoplasma pneumoniae (Fast Track Diagnostics, Esch-sur-Alzette, Luxembourg). None of the listed pathogens could be detected.

The boy had not been vaccinated against SARS-CoV-2 and had no history of previous COVID-19 disease, and his serum tested negative for SARS-CoV-2-specific antibodies.

An abdominal ultrasound revealed a large conglomerate of lymph nodes in the right lower abdomen and enteritic bowel loops (Figure 1a). The increased blood circulation within the lymph nodes was visualized by Doppler ultrasound (Figure 1b). The appendix was not visible. Subsequent magnetic resonance imaging (MRI) of the lower abdomen showed a significant thickening of the intestinal wall in the terminal ileum and the ileocecal transition, indicative of terminal ileitis together with remarkably extensive abdominal lymphadenopathy, with the appendix appearing normal (Figure 1c–f).

### 2.4. Treatment and Progress

Symptomatic treatment consisted of intravenous fluid administration and pain management. Given the absence of leukocytosis, a negative procalcitonin, and a favorable response to symptomatic therapy, and despite markedly elevated C-reactive protein levels, antibiotics were not administered. A follow-up ultrasound the next day showed an unchanged condition. Inflammatory parameters improved and abdominal symptoms resolved. The patient was discharged from the hospital after three days, fever-free and with minimal residual complaints. The boy recovered completely and has remained free of gastrointestinal symptoms for more than 12 months. Resolution of the thickened wall of the terminal ileum and of the lymphadenopathy was demonstrated by a follow-up ultrasound.

## 3. Case 2

### 3.1. History

A 16-month-old patient presented with a fever of up to 40 °C and bloody diarrhea for one day. The patient had no pre-existing medical conditions. But three weeks earlier, the infant had already gone through a short illness with fever and diarrhea, without respiratory symptoms. At the same time, the mother had a flu-like illness.

### 3.2. Clinical Findings

Upon admission, the patient had tachycardia (170/min), normal blood pressure, and normal body temperature. Physical examination was challenging due to the patient’s resistance, but no abdominal guarding was observed.

### 3.3. Diagnostic Evaluation

The laboratory results showed a significantly elevated CRP level (12.2 mg/dL), with only a slight elevation in procalcitonin (1.6 ng/mL). Blood counts, differential counts, and electrolytes were normal. Venous blood gas analysis revealed a base excess of −7.1 mmol/L (normal range from 2.0 to 3.0 mmol/L) and a bicarbonate level of 19.6 mmol/L (normal range from 21.0 to 26.0 mmol/L), with a balanced pH.

An abdominal ultrasound revealed an enteritic pattern with fluid-filled bowel loops. The bowel wall of the cecum was significantly thickened to 5 mm, along with swelling of the ileocecal valve and the wall of the terminal ileum, indicative of terminal ileitis and cecitis (Figure 2a–c).

SARS-CoV-2 nucleic acid was detected at the limit of detection in the first throat swab (cycle threshold of 43). A second throat swab remained positive with a cycle threshold value of 31.9, compatible with a recent SARS-CoV-2 infection. A stool sample also tested positive, with the XBB1.9 Omicron subvariant identified in both the stool and throat swabs. 

However, Campylobacter jejuni was identified in the stool culture on a Campylobacter blood-free selective agar (Oxoid, Thermo Fisher Diagnostics, Austria GmbH, Vienna, Austria). The remaining extensive screening, as in case 1, for other viral and bacterial pathogens was negative. 

The infant’s father also had symptoms of watery diarrhea, and Campylobacter jejuni was cultured from his stool too.

### 3.4. Treatment and Progress

The patient received symptomatic treatment with intravenous rehydration and subsequent continuous intravenous fluid therapy over five days due to persistent overall discomfort and reduced oral fluid intake. Without antibiotics, the fever declined spontaneously, and laboratory infection parameters improved. Mucus and occasionally bloody diarrhea persisted during hospitalization.

## 4. Discussion

We report on two pediatric cases of acute terminal ileitis associated with SARS-CoV-2 infection. One of them appears to be a rare case of isolated terminal ileitis caused exclusively by acute SARS-CoV-2 infection. Only a few similar cases in children have already been reported. In 2020, a cluster of eight children were admitted to Great Ormond Street within 8 days for suspected appendicitis, which turned out to be terminal ileitis with documented SARS-CoV-2 infection [24]. A four-year-old boy in this series probably had isolated terminal ileitis. Four others suffered from severe systemic inflammatory syndrome in addition. Three further cases in this series had negative SARS-CoV-2 tests. Two additional single-case reports described COVID-19-associated isolated terminal ileitis without respiratory symptoms or systemic symptoms: The first, in 2021, was claimed by the authors to be the first reported case of isolated terminal ileitis, occurring in a 48-year-old woman [25]. The second case involved a 29-year-old man [26]. However, in the context of MIS-C, gastrointestinal manifestations are common. A study on gastrointestinal disease in 35 children with MIS-C demonstrated that 34 (97%) had gastrointestinal complaints, and 4 (11%) showed marked terminal ileitis (4 out of 7 patients (57%) who had abdominal CT scans) [27]. There are also single-case reports on complicated terminal ileitis in MIS-C [28,29]. In a recent retrospective study on acute terminal ileitis from a Turkish pediatric emergency department, 143 patients with terminal ileitis were identified among 5363 patients who required abdominal imaging, and MIS-C was one of the top three identified causes [30].

The histological picture of SARS-CoV-2-associated terminal ileitis was documented in a nine-year-old girl with MIS-C and suspected appendicitis who was treated through ileocecal resection [31]. Histopathology revealed transmural inflammation and extensive subserosal edema of the ileum, mesenteric necrotizing lymphadenitis, and vasculitis of the mesenteric arteries and veins. Histological reports of isolated terminal ileitis from the acute phase of the SARS-CoV-2 infection are not available.

Most cases of severe gastrointestinal manifestations described in the literature received a polypragmatic therapeutic regimen, including broad-spectrum antibiotics, specific antiviral therapies, IVIG, and circulatory support measures, up to surgical interventions [24,28,32,33,34,35].

The treatment of our patients consisted of supportive measures only, including intravenous fluid administration. Spontaneous improvement occurred in both presented cases. In the absence of leukocytosis and significant procalcitonin elevation, antibiotic therapy was avoided despite high CRP levels.

COVID-19 in childhood presents with a heterogeneous and nonspecific clinical spectrum. Although it is considered primarily a respiratory disease, it may present as an acute abdomen without any respiratory symptoms [24,25]. Gastrointestinal symptoms associated with SARS-CoV-2 infection are common in childhood. They include a loss of appetite, nausea, vomiting, diarrhea, and abdominal pain. According to a review by Puoti M.G. et al. [36], the reported overall prevalence of gastrointestinal symptoms in pediatric COVID-19 patients is remarkable but highly variable. In these reviewed papers—we restricted the analysis to those with at least 100 patients—the overall prevalence of gastrointestinal symptoms ranged from 11% to 24%, and that of abdominal pain from 1.6% to 18.8% [36]. However, in children with MIS-C, gastrointestinal symptoms are a major component of the presentation, seen among 84%, with 75% experiencing abdominal pain [9]. Given this high prevalence, COVID-19 should be considered in the differential diagnosis when gastrointestinal symptoms occur in children. Apart from terminal ileitis and mesenteric lymphadenitis, both of which can clinically mimic acute appendicitis, hemorrhagic colitis is another severe gastrointestinal manifestation of COVID-19 that may precede respiratory symptoms by several days [37].

In a study by Yock-Corrales A. et al. [10] of 1010 pediatric patients with COVID-19 or MIS-C, only 42 children (4.2%) presented with an acute abdomen. In total, 38 patients underwent surgery, of which 34 (90%) were diagnosed with appendicitis, 2 with mesenteric lymphadenitis, and 2 had normal abdominal findings. No case of terminal ileitis was found in this large cohort, highlighting the rarity of this clinical presentation [10]. Extensive lymphadenopathy, as depicted in our first case, was reported earlier as a presenting initial manifestation of COVID-19, which progressed to severe disease requiring intensive care [38]. Prominent mesenteric lymphadenopathy was also described in the series by Tullie L. et al. [24]; it appears to be a rare or underreported but characteristic gastrointestinal manifestation of COVID-19.

Infectious ileocecitis is a common cause of pain in the lower right quadrant of the abdomen. It mimics acute appendicitis and is an important differential diagnosis. Traditionally, it is caused by infection with Yersinia, Campylobacter, or Salmonella. Importantly, these three pathogens present with a distinct pattern of ileocecal infection. Salmonella typically affects the cecum and the ascending colon, Campylobacter jejuni often causes moderate terminal ileitis in addition, while prominent terminal ileitis together with pronounced mesenteric lymphadenopathy is a characteristic feature of Yersinia enteritis [39]. SARS-CoV-2 is a new pathogen causing terminal ileitis. The pattern of ileocecal infection is similar to that of Yersinia, as demonstrated in our case 1 and by Tullie L. et al. [24].

SARS-CoV-2 gains entry into host cells by binding the viral spike (S) glycoproteins to the angiotensin-converting enzyme-2 (ACE-2) receptor, which is highly expressed on type I and II alveolar epithelial cells in the lungs. However, it is also found in many extrapulmonary tissues, such as the gastrointestinal tract, heart, liver, and kidneys. High levels of ACE-2 receptors are present on the luminal surface of differentiated epithelial cells in the small intestine. The primary mode of gastrointestinal tract involvement, whether through fecal–oral transmission or secondary to respiratory infection, has not been definitively established [36]. The exclusive intestinal manifestation, as seen in our patient, suggests the former. In addition, a study by Xu Y. et al. [40] showed viral shedding in stool samples for up to 5 weeks after the first negative nasopharyngeal aspirate in children, indicating a potentially significant route of transmission. 

SARS-CoV-2 has a significant impact on the gut microbiome, causing alterations in its composition, often leading to dysbiosis. Infected individuals may experience a reduction in beneficial bacterial species and an increase in opportunistic pathogens. These changes are attributed to the body’s immune response, inflammation, and possible antibiotic use during the infection [41]. The study by Brogna et al. explores the possibility that SARS-CoV-2 might exhibit bacteriophage-like behavior. They found indications that the virus might interact with and possibly infect gut bacteria, suggesting a novel interaction that could influence the understanding of the impact of COVID-19 on the gut microbiome and potentially inform new treatment strategies [42].

Research is also ongoing to determine whether COVID-19 acts as a risk factor or trig-ger for chronic IBD. COVID-19 causes immune dysregulation and an exaggerated systemic immune response. IBD results from a dysregulated or inappropriate immune response to environmental factors in conjunction with a genetic predisposition. It is therefore plausible that SARS-CoV-2 could be a potential trigger for IBD. There are already some case reports in the literature of newly developed IBD in adolescents during or after COVID-19 [43].

Finally, our second case emphasizes the need for a complete microbiological workup for additional gastrointestinal pathogens, even with evidence of SARS-CoV-2. In this case, coinfection with Campylobacter jejuni was detected. The exact role of each pathogen in the disease remains speculative. Coinfections of Campylobacter jejuni and coronavirus-like particles were described in primates in 1985 by Russell R. et al. [44]. They may also occur in humans, as shown in our second case and in a large Italian study where five (3.4%) Campylobacter coinfections were documented in 685 children with COVID-19 [45].

## Figures and Tables

**Figure 1 microorganisms-12-01377-f001:**
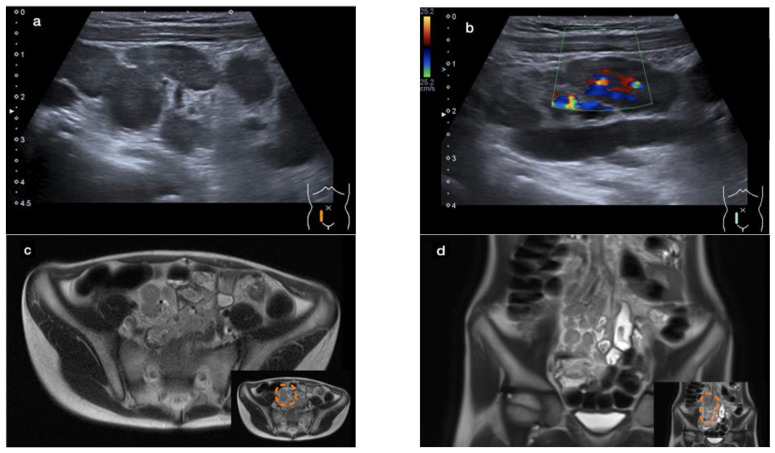
Sonographic imaging of a large conglomerate of lymph nodes with increased blood flow shown by a Color Doppler ultrasound in the right lower abdomen (**a**,**b**). MRI showing extensive lymphadenopathy in the right lower and middle abdomen (**c**,**d**) and the thickened, diffusion-impaired terminal ileum (**e**,**f**), as highlighted within the orange dashed line in the lower right corner of the images (case 1).

**Figure 2 microorganisms-12-01377-f002:**
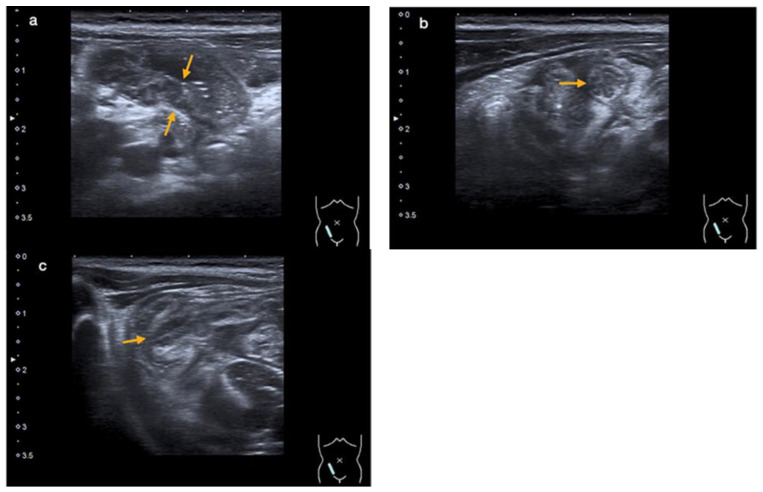
An abdominal ultrasound showing thickening of the intestinal wall of the terminal ileum (orange arrows in (**a**)) and the ileocecal valve (orange arrow in (**b**)—cross-section, and (**c**)—longitudinal section) (case 2).

## Data Availability

The original contributions presented in this study are included in the article; further inquiries can be directed to the corresponding author.

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
