# Peer review of "Terminal Ileitis as the Exclusive Manifestation of COVID-19 in Children"

_microorganisms, 2024, doi:10.3390/microorganisms12071377_

Round 1
Reviewer 1 Report
Comments and Suggestions for Authors
The authors' work, is a brief commentary of two case reports of two children 16 months and 9 years old, one associated with campylobacter co-infection.
It is an important work that needs more detailed and expanded.
In the history of coronaviruses in general and in the scientific medical literature, some studies point out the presence of campylobacter with coronaviruses N63 and 299E. The authors should find these papers from the 1970s and 1980s and include them in the discussion because they can help future research. In addition, the bacteriophage presence of this virus now seems certain, and the authors should also consider this when discussing gastrointestinal symptoms.
The materials and methods must be better detailed when discussing the nanopore extraction and analysis method.
Author Response
Dear Reviewer 1,
thank you for your helpful comments on our manuscript “Terminal Ileitis as a Manifestation of COVID-19 in Children”. Your feedback has been included in the manuscript and is summarised in this letter:
- In the history of coronaviruses in general and in the scientific medical literature, some studies point out the presence of campylobacter with coronaviruses N63 and 299E. The authors should find these papers from the 1970s and 1980s and include them in the discussion because they can help future research.
Unfortunately an extensive search for the suggested publications with coinfection of coronaviruses and campylobacter revealed only one report on primates. An Italian study documented 5 campylobacter coinfections in children with COVID-19.
Therefore the authors discuss these two publications [44 and 45] between lines 281-284.
[44] Russell RG, Brian DA, Lenhard A, et al.. Coronavirus-like particles and Campylobacter in marmosets with diarrhea and colitis. Dig Dis Sci. 1985 Dec;30(12 Suppl)-72S-77S.
[45] Lo Vecchio, A.; Garazzino, S.; Smarrazzo, A.; Venturini, E.; Poeta, M.; Berlese, P.; Denina, M.; Meini, A.; Bosis, S.; Galli, L.; et al. Factors Associated With Severe Gastrointestinal Diagnoses in Children With SARS-CoV-2 Infection or Multisystem Inflammatory Syndrome. JAMA Netw Open 2021, 4, e2139974.
- In addition, the bacteriophage presence of this virus now seems certain, and the authors should also consider this when discussing gastrointestinal symptoms.
The authors have taken up this point and discuss this topic in a new paragraph (lines 262-270) adding two important publications.
[41] Zhou, B.; Pang, X.; Wu, J.; Liu, T.; Wang, B.; Cao, H. Gut microbiota in COVID-19: new insights from inside. Gut Microbes 2023, 15, 2201157.
[42] Brogna C, Costanzo V, et al. Analysis of Bacteriophage Behavior of a Human RNA Virus, SARS-CoV-2, through the Integrated Approach of Immunofluorescence Microscopy, Proteomics and D-Amino Acid Quantification. Int J Mol Sci. 2023 Feb 15;24(4)-3929.
- The materials and methods must be better detailed when discussing the nanopore extraction and analysis method.
The authors added this information (see lines 92-101)

Reviewer 2 Report
Comments and Suggestions for Authors
The manuscript and the clinical findings findings are interesting and well documented. Especially, the post pandemic SARS-CoV-2 era this manifestation should be considered with every other possible infection it might cause. It will not only improve the clinical studies also it may provide a better and adequate treatment regimen to the patients. Especially, considering the route of entry of the virus it is really important to account the effect the infection with other gastrointestinal microbiological manifestations.Terminal ileitis, being a rarity in a large cohort of 1010 pediatric patients with COVID-19 or MIS-C as described the previously published article makes this clinical finding even more exciting to consider this manuscript as a good report of the characteristic gastrointestinal manifestation of asymptomatic COVID-19 in children.
I suggest only some minor comments to be considered.
Line no. 88-89 and 203-204: Digits seem to have commas instead of points.
Line 147-149: The sentence may be reconstructed in a meaningful way.
Author Response
Open Review
(x) I would not like to sign my review report
( ) I would like to sign my review report
Quality of English Language
( ) I am not qualified to assess the quality of English in this paper
( ) English very difficult to understand/incomprehensible
( ) Extensive editing of English language required
( ) Moderate editing of English language required
( ) Minor editing of English language required
(x) English language fine. No issues detected
|
Yes |
Can be improved |
Must be improved |
Not applicable |
|
|
Does the introduction provide sufficient background and include all relevant references? |
(x) |
( ) |
( ) |
( ) |
|
Is the research design appropriate? |
(x) |
( ) |
( ) |
( ) |
|
Are the methods adequately described? |
(x) |
( ) |
( ) |
( ) |
|
Are the results clearly presented? |
(x) |
( ) |
( ) |
( ) |
|
Are the conclusions supported by the results? |
(x) |
( ) |
( ) |
( ) |
Comments and Suggestions for Authors
The manuscript and the clinical findings findings are interesting and well documented. Especially, the post pandemic SARS-CoV-2 era this manifestation should be considered with every other possible infection it might cause. It will not only improve the clinical studies also it may provide a better and adequate treatment regimen to the patients. Especially, considering the route of entry of the virus it is really important to account the effect the infection with other gastrointestinal microbiological manifestations.Terminal ileitis, being a rarity in a large cohort of 1010 pediatric patients with COVID-19 or MIS-C as described the previously published article makes this clinical finding even more exciting to consider this manuscript as a good report of the characteristic gastrointestinal manifestation of asymptomatic COVID-19 in children.
I suggest only some minor comments to be considered.
Line no. 88-89 and 203-204: Digits seem to have commas instead of points.
Line 147-149: The sentence may be reconstructed in a meaningful way.
June, 27th 2024
Dear Reviewer 2,
thank you for your helpful comments on our manuscript “Terminal Ileitis as a Manifestation of COVID-19 in Children”. Your feedback has been included in the manuscript and is summarised in this letter:
- Line no. 88-89 and 203-204: Digits seem to have commas instead of points.
Thank you for these details: the authors removed the commas in lines 88-89 and 223-224.
- Line 147-149: The sentence may be reconstructed in a meaningful way.
The authors have rewritten this statement (see lines 154-157).

Reviewer 3 Report
Comments and Suggestions for Authors
The manuscript entitled "Terminal Ileitis as the exclusive Manifestation of COVID-19 in Children" is interesting, and is an hot topic. However, there are several issues need to be improved.
1. The abstract need to be summarized because now it is too much long. The abstract should focus on the overall content of the article, and describe the clinical performance and conclusion of the case.
2. Two cases were found in 2023, but the cases cited in the author's manuscript were partly in 2020 and previous years. It is recommended to find the literature and update similar cases from 2020 to make them more convincing.
3. There are also some spelling and formatting errors throughout the manuscript. Therefore, the authors should thoroughly and carefully examine the language and formatting errors.
4. Authors should re-examine the references, including the correct reference format, citation order, number, etc., and, if necessary, recommend reference to the format of the latest issue of the journal.
Comments on the Quality of English LanguageMinor editing of English language required
Author Response
Open Review
(x) I would not like to sign my review report
( ) I would like to sign my review report
Quality of English Language
( ) I am not qualified to assess the quality of English in this paper
( ) English very difficult to understand/incomprehensible
( ) Extensive editing of English language required
( ) Moderate editing of English language required
(x) Minor editing of English language required
( ) English language fine. No issues detected
|
Yes |
Can be improved |
Must be improved |
Not applicable |
|
|
Does the introduction provide sufficient background and include all relevant references? |
( ) |
(x) |
( ) |
( ) |
|
Is the research design appropriate? |
( ) |
(x) |
( ) |
( ) |
|
Are the methods adequately described? |
( ) |
(x) |
( ) |
( ) |
|
Are the results clearly presented? |
( ) |
(x) |
( ) |
( ) |
|
Are the conclusions supported by the results? |
( ) |
(x) |
( ) |
( ) |
Comments and Suggestions for Authors
The manuscript entitled "Terminal Ileitis as the exclusive Manifestation of COVID-19 in Children" is interesting, and is an hot topic. However, there are several issues need to be improved.
- The abstract need to be summarized because now it is too much long. The abstract should focus on the overall content of the article, and describe the clinical performance and conclusion of the case.
- Two cases were found in 2023, but the cases cited in the author's manuscript were partly in 2020 and previous years. It is recommended to find the literature and update similar cases from 2020 to make them more convincing.
- There are also some spelling and formatting errors throughout the manuscript. Therefore, the authors should thoroughly and carefully examine the language and formatting errors.
- Authors should re-examine the references, including the correct reference format, citation order, number, etc., and, if necessary, recommend reference to the format of the latest issue of the journal.
Comments on the Quality of English Language
Minor editing of English language required
June, 27th 2024
Dear Reviewer 3,
thank you for your helpful comments on our manuscript “Terminal Ileitis as a Manifestation of COVID-19 in Children”. Your feedback has been included in the manuscript and is summarised in this letter:
- The abstract need to be summarized because now it is too much long. The abstract should focus on the overall content of the article, and describe the clinical performance and conclusion of the case.
The authors have attempted to summarize this manuscript in 191 words in the abstract. Authors would not shorten this length as it meets the requirements of the journal. The clinical presentation is mentioned in lines 20/21. Imaging studies are described in lines 22/23 and the conclusions can be found in lines 25-29.
- Two cases were found in 2023, but the cases cited in the author's manuscript were partly in 2020 and previous years. It is recommended to find the literature and update similar cases from 2020 to make them more convincing.
It is a good idea to expand this aspect with new literature (Qanneta 2023, Sahn 2021, Fivelman 2022, Koyuncu 2024) (see lines 189 to 200).
- There are also some spelling and formatting errors throughout the manuscript. Therefore, the authors should thoroughly and carefully examine the language and formatting errors.
The following corrections have been made:
Lines 88-89; 223-224: The authors have removed commas and put full stops instead.
Line 124: Color
Line 183: a cluster of eight children was…
- Authors should re-examine the references, including the correct reference format, citation order, number, etc., and, if necessary, recommend reference to the format of the latest issue of the journal.
The authors have adopted the citation style and corrected the reference format for example all DOIs have been removed.
Comments on the Quality of English Language: Minor editing of English language required
The following corrections have been made:
Line 109: and
Lines 154-157: The authors have rewritten this statement.
Line 180: acute terminal ileitis
Line 207: received
Line 228: …, which both can clinically mimic
Line 251: SARS-CoV-2 gains entry into host cells by binding of….

Round 2
Reviewer 1 Report
Comments and Suggestions for Authors